# Rational Organization of Urban Parking Using Microsimulation

Irina Makarova [1,*], Vadim Mavrin [1], Damir Sadreev [1], Polina Buyvol [1,*], Aleksey Boyko [1] and Eduard Belyaev [1,2]

1   Naberezhnye Chelny Institute, Kazan Federal University, Syuyumbike Prosp. 10a, 423822 Naberezhnye Chelny, Russia
2   Institute of Digital Technologies and Economics, Kazan State Power Engineering University, Str. 2nd South West, 26, 420034 Kazan, Russia
*   Correspondence: kamivm@mail.ru (I.M.); skyeyes@mail.ru (P.B.)

**Abstract:** Urbanization, which causes the need for population mobility, leads to an increase in motorization and related problems: the organization of parking spaces in cities, both near work places and recreational spaces, and not far from residential locations. This has a number of consequences. Therefore, the occupation of parking spaces near shopping centers and sports and recreation facilities, intended only for customers of these organizations, makes it difficult for direct customers to access services. This forces potential customers to look for a parking space in adjacent areas, often far from the target location. At the same time, the search for a parking space is stretched over time, negatively affecting the environment in the form of emissions and noise. On the other hand, there is a risk of losing a client. In the course of the study, we have analyzed the state of the problem and the directions of research on parking management in cities, and then we have studied the possibilities of using simulation models to find rational options for the organization of access to parking spaces and further using such models in decision support systems (DSS) as an intellectual core. The literature review showed that this is the most adequate option for an intelligent city parking space management system. At the same time, the environmental factor must also be taken into account. Research methods are based on field studies of traffic flows and emissions near parking places, and mathematical and simulation modeling. The proposed system will allow the evaluation of the effectiveness of the proposed changes in the organization of access to parking spaces, and, in the future, when implementing the obtained optimal solution, in practice, provide customers with a guaranteed parking space and reduce traffic and emissions. The introduction of such a system guarantees its quick payback, which is associated with the efficiency of use, as well as with the additional effects obtained from its implementation (improving the road situation, reducing vehicle emissions, solving social problems of the population, etc.), which is especially important for medium and small cities with limited budgets.

**Keywords:** parking management systems; smart parking; simulation models; decision support systems

## 1. Introduction

One of the modern city problems, which is aggravated by the increase in the motorization level, is the lack of parking spaces, which refers both to places where short-term demand is formed (near business, cultural, shopping, entertainment centers) and to population residence places (long-term vehicle storage). This problem has a number of consequences associated with the formation of spontaneous parking lots that impede the transport traffic, leading to traffic jams and congestion, as well as worsening the environmental situation in cities [1–3]. The issue of parking management is solved by creating parking management systems. However, there are a number of unresolved issues that are typical for medium and small cities with small budgets: implementation of such a system should guarantee its quick payback, which is associated with the efficiency of use,

as well as with the effects obtained from the solution's implementation (road situation improvement, reducing vehicle emissions, solving social population problems, etc.).

In the city of Naberezhnye Chelny (which we study), parking spaces near places of residence are free, but their number becomes insufficient with the increase in the motorization level. In this regard, residents occupy places near the centers of the formation of short-term demand. Therefore, there is a problem of a lack of parking spaces near business, shopping, entertainment and sports centers, which are designed based on the number of possible customers in accordance with urban planning standards based on the number of parking spaces per given area of the facility. In large cities and metropolitan areas, this problem is partially solved by organizing paid parking lots. In medium and small settlements, this strategy turns out to be ineffective, since the level of average income is not high enough and such places will not be in demand, which means that there is a risk of losing potential customers. Currently, if these places are occupied by the clients of other places of attraction or residents of nearby houses, the clients of the object in question experience inconvenience and are forced to look for a parking space in adjacent areas, often far from the target location. Therefore, it seems logical to restrict access to parking spaces by dividing the flow into "own" and "another". In this case, a difficulty arises—some customers arrive by taxi, and there is a need to account for such short-term arrivals. In any case, the implementation of the complex logic of organizing access to the parking space should be supported by a decision support methodology, as well as taking into account environmental aspects.

Therefore, the main goal of the article was to study aspects of the rational organization of parking spaces in small and medium-sized towns, taking into account the environmental factor. To do this, we identified four research questions:

1. What are the possible ways to organize efficient parking spaces today?
2. What is the structure of the decision support system for managing the organization of parking spaces?
3. How can we effectively consider the environmental factor when choosing the location of the parking space?
4. What methods are most effective in organizing parking spaces near public places of trade, sports and recreation in small and medium-sized towns?

The study resulted in the developed concept of a decision support system. The intellectual core of it is a simulation model that allows the consideration of various options for organizing access to parking spaces. The objectives and questions of the study determined the structure of the article. Thus, the second section provides an overview of the articles currently available in scientometric databases on the methods and models of organizing effective parking spaces today. The third section substantiates the expediency of using the method of simulation micromodeling in the study of a parking space and describes the method of preliminary ecological analysis of the territory. The fourth section is devoted to the development of the concept of a decision support system, the practical implementation of the method for assessing the ecological analysis of the territory of the city under consideration, as well as the construction of a simulation model of a parking space near a sports facility and a description of a computer experiment. The fifth section includes a generalized discussion. The sixth section is devoted to results and further research.

## 2. Literature Review: Existing Parking Management Methods

### 2.1. Forecasting and Managing Parking Demand

Most cities have limited parking spaces, and, since it is difficult and costly to create new parking spaces, it is imperative to make the most of existing parking spaces. The article [4] presents a city parking management system based on equal access to the parking infrastructure for any road user. The method is based on the use of three ensemble regressors. The authors proposed a combination of models to predict the number of vacancies in a given area at a specific time of day.

Research [5] examines traffic, including in relation to parking regulations. The authors investigate traffic in the city of Uherske Hradiste near specific parking spaces, including

paid parking zones, especially designated parking spaces near institutions and for time-limited free parking, in order to develop measures and proposals to improve the efficiency of the city's transport system within the framework of the Smart City concept.

Since finding a parking space has become increasingly difficult and expensive lately, the authors [6] propose a balancing mechanism for efficiently sharing both public and private parking lots (PL). The problem of distributing parking lots contains two goals: (1) minimization of parking costs and (2) balancing the demand for parking among several PLs. The authors use an algorithm based on the variable direction multiplier method, which can provide a distributed implementation and has a positive effect. In addition, due to the growing fleet of electric vehicles, the authors plan to use a balanced demand mechanism to balance electricity congestion between multiple charging stations.

Article [7] describes the booking methods used in the field of intelligent parking management systems. This allowed us to define three steps required for a booking system: discovery, optimization and assignment. The authors highlighted the most important features of a high-performance booking system: taking into account driver preferences and optimizing distances and delays in selection, which is necessary to ensure the scalability and stability of the system.

The article [8] proposes a distributed structure for allocating parking spaces based on the adaptive pricing algorithm, hashgraph consensus and virtual voting. Using the model, all users and parking owners can easily agree on the allocation of a parking space using the minimum bandwidth. In addition, an adaptive pricing model is proposed to increase the total income of parking owners and the convenience of users. The simulation results show that the proposed model is very useful for saving the user's time in searching for free parking spaces, avoiding congestion and making optimal use of resources.

The article [9] proposes a new method of parking space management for shared vehicles based on open big data available, including the visualization of GPS order data and analysis of data distribution on the road. The authors apply a clustering algorithm using GPS data for clustering. The optimization algorithm is used to set the sum of the distances from the location of all orders to the closest stopping point as the optimization target, optimize the location of parking spaces and reduce the walking distance of customers.

The article [10] proposed short-term methods for predicting the available parking space. The results showed that compared to the most used approaches, the stability and accuracy of predictions are significantly improved, although there are limitations in the form of an unsatisfactory waste of time.

Thus, we can distinguish two vectors of modern research in the field of parking management: the prediction of the occupancy of free parking spaces and the organization of paid parking zones and their impact on nearby traffic.

## 2.2. Models and Algorithms for Finding Vacant Places for Smart Parking

One of the main reasons for the high traffic congestion in cities is the disorganized search for free parking spaces. Thus, the results obtained by Bischoff J. and Nagel K. [11] show that parking search traffic constitutes up to 20% of the total traffic in a residential area and has a significant impact on the total travel time of agents traveling by vehicle. This leads to financial and environmental problems. The allocation of parking lots is a dynamic task since the input data are constantly updated. The article [12] proposes a method to solve the problems of updates in real time using a greedy heuristic, which, according to the authors, is rational if there is enough space in the parking lot to accommodate all vehicles. However, when there is a decrease in capacity or a large number of vehicles, the exact algorithm is the best choice. To provide a more accurate distribution, especially when the vehicle speed is not considered constant, the authors propose to include a double horizon heuristic that will track the effect of the current decision and adapt future ones.

A study of various types of intelligent parking systems has revealed a number of problems that are associated with the serious and costly modernization of parking spaces, such as a wired power supply or regular battery replacement. The proposed system [13]

can control up to four parking spaces from one node, transmits data via a wireless network and uses solar panels for operation, which does not require maintenance and ensures environmental friendliness. The authors propose an application for the phone, which, using Google Maps, can indicate routes on a map to a destination.

The article [14] proposes a structure based on the adaptation of the "day–night" domain; the key idea of the framework is to embed images in two spaces. Taking advantage of the two-domain exchange, the framework not only transfers knowledge and tags between domains, but also synthesizes images, which allows the parking system to more efficiently determine the status of places at night. The proposed framework will help to extend this parking inference system to new environments with reduced data collection and tagging costs.

Vehicle drivers spend a lot of time looking for free spaces in multistory vehicle parks, which creates queues and congestion on the roads. In the study [15], an intelligent parking system based on image processing was developed for multi-storey garages. The proposed system design using Python IDLE and the OpenCV library uses combined edge detection and anchored pixel coordinates to determine if a parking space is occupied in the resulting footage.

The paper [16] proposes a research methodology for empirically measuring the impact of street parking policies based on automated parking transactions for visitors in central Stockholm, Sweden. As the authors have found, the average occupancy rate does not reflect the ease of finding a free parking space, and this should be taken into account when predicting the consequences for cruise traffic. The process of finding parking on the street, according to the authors, can be simplified by considering a sequence of independent Bernoulli tests with a failure rate corresponding to the average parking load in a given area.

The explicit parking search is not as widely integrated into the transport's modeling and transport models. The article [11] demonstrates the integration of parking search simulation in MATSim (Multi-Agent Transport Simulation). This includes integration into agent modeling logic using the day's rescheduling methodology, splitting vehicle trips into multiple segments for each leg of the trip, parking search behavior and a data structure for parking infrastructure. Research results for a district in Berlin show that the average time that it takes to find parking and walk to the actual destination (or back to the vehicle) in this area is 8 min. Further research should include the impact of different parking search strategies and their impact on travel times.

Finding a parking spot leads to congestion, increased air pollution and negative emotions. In the article [17], the authors propose to determine the congestion of open parking spaces using real-time vehicles with a thermal imager and deep learning, believing that such information will help to reduce congestion and subsequent air pollution. A thermal camera is used to collect videos in various environmental conditions, and frames are extracted from these videos to prepare the dataset. The authors investigate deep learning networks (Yolo, Yolo-conv, GoogleNet, ReNet18 and ResNet50) for vehicle detection. The Yolo fast detector has been modified using convolutional and residual layers, which has hardly improved its performance. The use of filters can improve the performance of the detectors, but also increases the computational time of the detector, which is not suitable for real-time parking occupancy detection.

The article [18] presents a solution using smartphones for intelligent parking (ParkUs), with the provision of information about the parking availability at the selected destination. It is a crowdsourced approach based on mobile device detection and automatic part tagging of the user's journey with cruise/non-cruise events. This study's results show that reducing the search time for a parking place by even a few seconds will result in improved air quality and significant reductions in $CO_2$ emissions, which is the rationale for using ParkUs.

The document [19] proposes an application by which the driver selects a preferred parking spot, tracks the elapsed time and uses electronic payments to avoid monetary transactions between staff. The quick response (QR) code is the key to reserving a parking space, confirming reservations and payments, as well entering and exiting. This system is convenient because it will provide non-cash transactions, and reduce interactions between



people, congestion in the parking lots and paper waste. In addition, the monitoring system increases the safety of parked vehicles.

The article [20] introduces the Smart Car Parking System, which will help car owners to quickly find parking spots using an Android-based application system that stores both driver and parking provider information in a wireless database. The system concept includes both hardware and software for building the system. The system is integrated with a hardware solution using sensors; the LCD display is controlled by a microcontroller. However, there is room for improvement, especially with regard to security measures. To ensure security, a QR scanner, namely a VLP scanner, is involved.

The study [21] proposed an algorithm for extracting data from parking space sensors. Experiments and results show that the accuracy of the proposed algorithm can reduce the number of errors by retaining the total sleep time.

The Raspberry Pi system proposed in article [22] addresses the problem of existing smart parking systems using microcontrollers by developing a new application for an Android mobile phone. This allows any number of users to find a parking lot without registration due to vehicle detection units and the transmission of information about the availability of a parking space on each floor of the parking lot. This system can be implemented for shopping centers, buildings and cities in real time as a multi-level parking system, where the driver will park manually.

Study [23] describes the architecture of an intelligent parking management system with a wireless sensor network. Since the system is implemented using very low-cost devices, this can reduce the development costs.

The authors [24] propose an intelligent parking solution using wireless radio technology to locate a parked vehicle and transmit data from sensors to a central control system. The proposed system uses a self-forming network of dual-mode Bluetooth sensors in the parking lot. The localization technique is based on radio fingerprints using received signal strength indicator (RSSI) values from a beacon and a random forestry machine learning classifier that predicts where the vehicle is parked. The implementation uses Python on common Internet of Things (IoT) hardware, which allows for a variety of parking applications.

The authors of [25] are developing an intelligent parking system based on the ZigBee wireless sensor network. Thanks to the online and offline interactive management of parking spaces, the utilization rate of parking spaces is increased, and the problem of urban parking chaos is effectively solved, which is of practical importance.

Internal electronic devices not only make tasks easier for administrators, but also make parking smart. Thus, article [26] proposes an inexpensive, highly efficient and easy-to-manage parking management system based on the Alibaba Cloud platform and machine learning, which allows one to count the number of vehicles entering the parking lot and to recognize license plates, which is not only convenient for management but also helps drivers to find parking spaces. The system implements voice control both for parking and through the application.

The article [27] proposes a received signal strength indicator (RSSI) approach for detecting available parking spaces. The system uses a scalable message queue telemetry protocol to ensure its security. The advantage of the proposed system is the possibility of remote real-time detection of free parking spaces using the developed mobile application, which significantly saves time and money for vehicle owners and allows the system to be used in several parking lots; vehicle data are stored in a central database.

The integration of several effective technologies, such as the Internet of Things (IoT), unmanned aerial vehicles (UAVs) and 5G communications, can simplify the monitoring and management of parking spaces. The article [28] proposes an intelligent parking system that uses multiple sensors to track the occupancy of parking spaces, as well as surveillance from a UAV to improve accuracy. The system uses a magnetic sensor and an ultrasonic distance sensor, using the UAV to detect and correctly predict the presence of a vehicle in a parking space, which is used to inform users of the location of the nearest unoccupied parking space.

Study [29] is another example of the use and development of drone technology, namely the use of a quadcopter for counting vehicles in an open parking lot. The researchers recommend using a template that needs to be placed on parking spaces to more accurately determine if parking spaces are occupied or not and eliminate additional false positives. We used an SSD mobile network as a model, but, according to the authors, there are better models in terms of speed and accuracy—for example, Yolo V2 and Faster RCNN.

The authors [30] proposed a semi-guided and multi-tasking learning environment for determining the status of parking spaces using a magnetic signal for solving practical problems in street parking, such as environmental noise, non-unified coordinates of magnetic sensors, signal variations due to vehicle type, sensor location and adjacent moving vehicles. In addition, the authors propose a multitasking module for exploring distinctive and generalized characteristics using information from both tagged and untagged data.

Real-time automatic detection of the occupied parking space with a high level of accuracy is a useful concept for realizing intelligent parking. The authors [31] proposed a new form of automatic parking space detection system based on the Laplacian Edge Detection method in order to recognize when parking spaces are occupied or when the vehicle is parked incorrectly.

Since the volume of exhaust gases increases proportionally with an increase in the time for which a vehicle searches for a free parking space, the researchers focused on the primary task of developing algorithms, methods, applications and devices for recognizing unoccupied parking spaces and building the shortest route for the driver to them. In this case, computer vision methods, a magnetic sensor and an ultrasonic distance sensor are used.

*2.3. Machine Learning Algorithms for Parking Management*

Article [32] discusses the creation of simulation software that can provide information on available parking spaces. Methods used include character recognition with the EAST text detector algorithm, vehicle detection with the Haar cascade classification algorithm and free parking space detection. The study presents a detector that uses functional text to detect vehicles in parking spaces. These three methods are then combined into a modeling system that uses the Python and OpenCV libraries as tools. For further development, according to the authors, the Internet of Things (IoT) can be implemented.

The article [33] presents a virtual intelligent parking system based on a signal request mechanism. The free parking space selection mechanism is based on the proximity principle, the path planning is based on Dijkstra's algorithm, and the deadlock conflict resolution method is based on the signal request mechanism. The authors introduce a new type of dynamic priority that effectively improves the efficiency of resolving problems caused by deadlock conflicts.

The study [34] expands the application of federated learning to parking management by proposing FedParking, where parking operators (PLO) co-train a long-term short-term memory model to estimate a parking space. The authors formulate the interaction between PLO and vehicles as a Stackelberg multiplayer game. Given the dynamic arrivals of vehicles and time-varying parking capacity constraints, a deep learning, multi-agent approach and a DRL approach are applied to achieve Stackelberg equilibrium in a distributed but privacy-safe manner.

Research [35] offers smart parking solutions using big data analytics and deep learning techniques based on a convolutional neural network (CNN). The authors propose a cost-effective and convenient parking ecosystem that uses computer vision and deep learning to guide the user to the nearest parking space using a mobile app. The main focus of the work is on reducing the model training time and describing the classification model that is used for vehicle detection. CNNs are used to build a supervised classification model that detects vehicles in a parking lot.

The authors [36] propose to apply machine learning to automatically detect free spaces in delimited parking spaces, followed by an extension to non-delimited parking spaces.

This approach requires fewer images to train the classifier, accepts non-rectangular images of variable size as input and can also be applied to non-delimited parking spaces.

The paper [37] proposes a low-power wide-area network (LPWAN)-based urban parking space monitoring system with a focus on system design, including software, hardware and algorithms. LoRa technology and NB-IoT technology provide an opportunity in the future to form a large-scale low-power wireless sensor network for urban smart parking. Compared with other wireless sensor networks, the proposed parking space monitoring system has obvious advantages for advancements in urban smart parking projects.

The article [38] proposes a model for the joint distribution of overnight parking spaces between a residential area and an adjacent business area based on shared parking in the adjacent areas. The research findings of this article can provide a method for planning and managing shared parking in the surrounding areas of major cities in China. Based on the concept of shared parking and its premise, this article proposes building a model for the total distribution of nighttime parking spaces and defines the process of building a model and solving it by checking the correctness of the model assumptions.

The authors [39] have developed an intelligent parking system based on positioning and navigation technology in an ultra-wideband range. The whole system consists of three modules: an indoor positioning module, consisting of a UWB intelligent positioning bracelet (tag) and a positioning base station; a parking and shopping center data management module; and the user's mobile phone application module. The iParking smart parking system proposed in this article has more advantages than the video induction and reverse vehicle search system, both in function and cost, which will improve the level of parking intelligence, helping to transform traditional parking into intelligent parking.

The Integrated Intelligent Parking System (ISPS) includes empty parking space detection and offers the shortest route to a location, focusing on minimizing time and reducing unnecessary trips. In the study [40], ISPS supports users with automatic parking management and information about vehicle fuel use, helping to reduce congestion, human effort and carbon emissions.

The article [41] proposes a parking distribution model for a smart city, which uses a resource processing module (RHM) and a custom application to provide smart parking services, taking into account such parameters as distance, cost, time and traffic. Since the system takes into account several factors, this solution provides a more accurate estimate of the time required to reach the goal and helps to minimize errors.

The article [42] proposes a new scheme for optimizing parking costs for long-distance autonomous parking: AVPark. The system selects a temporary checkpoint where the client can pick up the AV for the trip. The user leaves the AV at the drop-off point, and the AV independently determines the optimal parking, using AVPark, taking into account the cost of parking, fuel consumption and the distance to a free parking space, and also seeks to minimize the distance for both drivers and the AV (traveling there and back from the drop-off point to the vehicle park). In the future, there will be great interest in integrating AVPs with edge computing and the cloud to support IoT services.

The article [43] proposes a parking system for resource management in an environment of mixed automated and human-driven vehicles as a problem of dynamic resource allocation of mixed-integer linear programming at each decision point in order to minimize the total costs for users. Based on the tree representation of the decision space for matching pairs, the Monte Carlo tree search and some heuristics are combined to find the optimal matching order (parking users and resources) in a short time. This online resource allocation system can effectively solve the city-wide parking problem and greatly improve the quality of service provided to human drivers based on the interaction mechanism developed in the mixed environment of automated and human-driven vehicles.

The next stage of development is the intellectualization of parking resource management by using machine learning algorithms, including for automated vehicles.

*2.4. Smart Parking Management Systems*

The parking lot is currently managed by parking assistants. When drivers are looking for a parking place, many vehicles move around, causing congestion and wasting time. The document [44] presents a computer vision-based parking monitoring and management system that makes it easier for drivers to find an empty parking space. The HAAR cascade classifier method detects and counts parked vehicles and then compares them to the available parking spaces. The accuracy rate for detecting vehicles for this study is 90%, and the accuracy rate for detected vacant parking spaces is 80%.

The article [45] describes the development and implementation of a parking management system that solves the problems of parking complexity and complex vehicle management, where the administrator can check the parking space and vehicle information at any time, and the driver can reserve a parking space via keywords. At the same time, this article analyzes the complexity of various path navigation algorithms, taking into account the different requirements for a real-time map in parking and off-site route navigation. For the parking space management module, this article analyzes Dijkstra's Heap Optimization Algorithm, Floyd's Algorithm and the Ant Colony Algorithm.

Sharing private parking spaces during idle periods has great potential to address urban traffic congestion and illegal parking in Smart Cities. The article [46] proposes a new privacy-enhancing parking sharing incentive scheme where parking space providers (PSPs) and customers are treated as buyers and dealers. At the same time, the distance between the customer's destination and the allocated parking space is minimized by solving a mixed-integer nonlinear programming problem, and the location confidentiality of customers' destinations is protected by the Laplace mechanism.

In the article [47], the authors point out that the growth in demand for parking requires the optimization of existing parking spaces. They propose the optimization of the parking of the sports center Enrique Razona in DLSU (Manila) using linear programming. Because its current access road is much wider than the minimum width required by the Philippines Building Code, it could be narrowed to make room for new parking spaces. This, in turn, will lead to reduced waiting times for drivers in the mornings and ultimately solve the parking problem at the university.

The article [48] proposes a new structure of a network model based on a recurrent neural network (RNN) and an ensemble learning algorithm called E-RNN. The ensemble learning algorithm model is used as the main learner, and the neural network is used as the secondary learner in ensemble learning. The new model parameters are optimized using the Particle Swarm Optimization (PSO) algorithm, which improves the model. Experimental results show that the algorithm has high accuracy and reliability for predicting parking spaces with small datasets.

The article [49] presents the structure of a blockchain-based parking management system designed to preserve the users' privacy without relying on a trusted third party. The proposed system combines BlockChainOpenSource (BCOS) and smart contract technology to share parking spaces.

The use of blockchain technology allows one to reduce intermediary costs through smart contracts. An important point in this ecosystem is the confirmation of participants' identities on the blockchain network. The article [50] proposes a blockchain-based collaborative mobility platform and demonstrates its validity. Based on ERC-721 tokens, a decentralized concept is created, implemented in a smart contract and supplemented by a hardware security module (HSM) to protect confidential key material. Finally, the system's evaluation by comparison with modern solutions has shown that the proposed solution has an advantage in security terms.

The main goal of researchers is to build an intelligent parking space management system, including the use of blockchain technology to guarantee security when sharing parking spaces. Certainly, a number of positive results have been obtained in this direction; however, an analysis of the areas of parking management research in cities did not reveal a similar example of solving the problem identified and described in the Introduction and

implementing a methodology to search for a parking space location, taking into account the environmental factor, and organizing the logic of access to it. The considered studies are aimed primarily at optimizing algorithms for finding free parking spaces. However, they do not consider a method to initially eradicate the problem by restricting access to the parking space. We did not find any attempts to introduce a complex logic of delimiting access to parking spaces, separating the flow of customers in time and selecting and estimating the parameters of this delimitation on a simulation model.

Thus, the study's goal was to develop the decision support system (DSS) concept and check the adequacy of the use of simulation micromodels and mathematical models in the information processing module as part of the DSS to manage the city parking spaces, taking into account the environmental factor.

### 3. Materials and Methods

DSSs are used to solve management problems in all areas of activity, especially when decision-making is based on processing big data. This allows one to create a multipurpose solution and systematize the methods and models used for collecting, analyzing and storing, as well as for processing, information. The use of simulation models (SM) as an intelligent core in a DSS will allow one not only to make informed decisions on parking area management based on forecasts, but also to choose the most effective decisions when it is necessary to change the number of parking spaces, both at the stage of existing parking lot reconstruction and during the new one's construction, as well as offering a way to organize access to the parking lot (Figure 1).

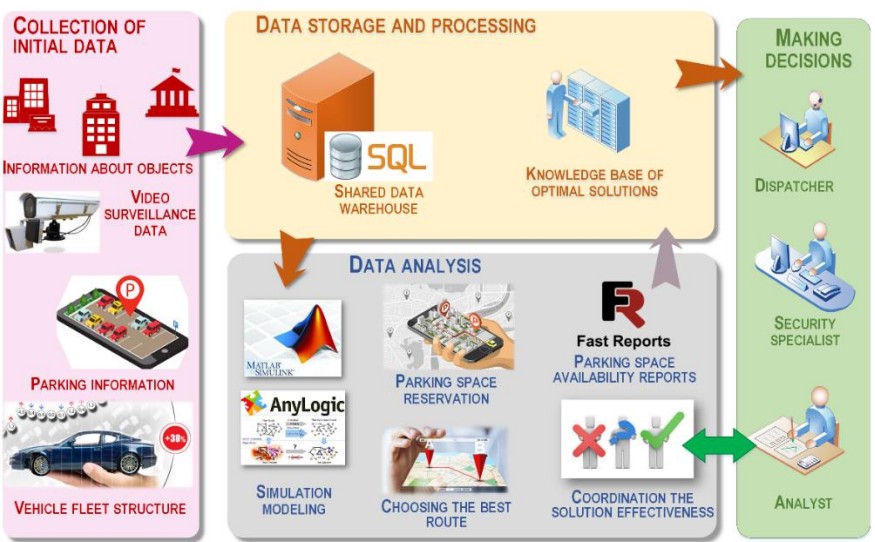

**Figure 1.** Conceptual structure of the DSS (drawn by authors).

Simulation has many advantages, such as relatively low costs, time savings, repeatability and visualization. Due to these qualities, modeling is one of the best methods of analysis and in searching for the most rational solution to various problems of road transport [51,52]. Thus, it is possible to simulate various road sections and parking spaces, determine their weak points and find solutions to improve traffic flow [53,54] and parking parameters, taking into account the various factors that have priority in each case.

We create simulation models using the AnyLogic software package. It contains a traffic library and a built-in optimizer. A micro-approach is used to simulate traffic flows in the AnyLogic environment. Models of this level make it possible to display the behavior of each individual road user who obeys the established rules, as well as their interactions, which makes it possible to study emergency situations and form a database of typical situations. At the same time, they take their place in the decision support system (Figure 1), which allows for informed decisions in critical situations to improve road safety. Using the

graphic elements of the AnyLogic environment, it is possible to set the number of traffic lanes, the width of the dividing lane, markings and the location of the STOP line. To set the route of the vehicles' movement, a process diagram is used within the framework of a discrete-event modeling approach. There is also an internal logic of vehicle behavior that does not contradict common sense and is implemented automatically at the level of the vehicle agent: choosing the vehicle's path taking into account speed limits, lane change logic, choosing a less busy lane, detecting (possible) collisions and taking measures to avoid them at intersections. An agent is a model element that can have behavior, memory (history), contacts, etc. In our case, the vehicle agent has the following characteristics: length, initial speed, preferred speed, maximum acceleration, maximum braking. It is also possible to set different vehicles' types with specific attributes and animations.

We conducted the study in the city of Naberezhnye Chelny. This is a small town with a population of 548,434 as of 2021. There are no electric vehicles in the city and there are no plans to distribute them. Instead of electric vehicles, compressed natural gas vehicles are gaining popularity. Refueling of these vehicles is carried out at combined gasoline–gas filling stations. In the fitness center, handicapped persons arrive in special cars or taxis that do not need parking, and this is a very small percentage.

To increase the efficiency of the use of parking spaces, in addition to dividing the flow of cars into "own" and "another", it is advisable to distinguish between clients by arrival time: for example, those clients who can visit the fitness center all day long are issued a black card, and those who visit only until 17.00 are issued a white card. Moreover, they need to be recognized and divided. This logic of organizing the fitness center's operations and, accordingly, the parking spaces must first be tested on a simulation model. Previously, we studied the occupancy of parking at different times of the day. We divided the working hours into 2 periods: from 7.:00 to 17:00 (standard working hours for the majority of the population) and 17:00 to 23:00 (standard non-working hours for the majority of the population). Further, for two days, we recorded at the time of the arrival and departure of vehicles in the parking lot, and whether the driver was a client of the fitness center or not. Thus, we were able to determine the average vehicle delay time in the parking lot (regardless of whether they were a client or not), the intensity of the vehicle' arrival and the share of vehicles that belonged to customers. Moreover, from the leadership of the fitness club, we obtained the ratio of black and white card holders.

The adequacy of the simulation micromodel is confirmed during the stages of verification and validation of the model. They are carried out on the basis of a comparison of the movement parameters of the road network section, calculated on the basis of data from the video recorder, with the corresponding parameters obtained as a result of the model run.

During the experiment on the model, the selected parameters of which will change in accordance with the plan of a multifactor experiment, it is possible to achieve the selected target functional—for example, the minimum vehicle driving time, volume of pollutant emissions, the degree of the parking space loading.

In general, the environmental factor is one of the decisive factors, both in the optimization of parking management and in the initial choice of its placement. Parking conditions are characterized by "volley" emissions of exhaust gases during launch and heating. The duration of this non-stationary regime, including heating of the cold engine, is 3–5 min in the warm season and 15–120 min in the cold season. At the same time, emissions of harmful substances, together with exhaust gases, are more than 8–10 times higher compared to stationary operation. In the case of a cold start, the vehicle emits CO by 86% more, SN—40%, $NO_X$—12% than during stationary operation [55]. Therefore, any decision on the creation of parking should be preceded by an environmental analysis of the territory.

Due to the fact that, as a rule, the main stationary sources of environmental pollution in urban areas are located outside the city or on its borders, motor transport makes the largest contribution to the city's environmental pollution. Therefore, for an ecological analysis of the city territory and determination of the places of the highest concentration of pollutants, it is necessary to calculate vehicle emissions based on well-known methods—

for example, methods for determining pollutant emissions into the atmospheric air from mobile sources—for conducting summary calculations of atmospheric air pollution [56].

Since emissions from stationary sources are conditionally constant, it is possible to construct scatter maps of pollutants, on which emissions from stationary sources will be presented as "background", based on the maximum permissible emissions and production control in compliance with environmental requirements. The calculation of emissions from traffic flows is performed by adding two components: emissions from a moving stream (1) and emissions from vehicles in the area of controlled intersections (2).

$$M_{L_i} = Lk_S \times \sum_i^k M_{k,i}^M \times G_k \times r_{vk,i}, \tag{1}$$

where

$L$ is the length of the UDS section, km;

$k_S = \frac{1}{3600}$ is the conversion factor of "hour" into "sec";

$M_{k,i}^M$ is the mileage emission of the $i$-th harmful substance by vehicles of the $k$-th group for urban operating conditions, g/km;

$k$ is the number of vehicle groups;

$G_k$ is the actual highest traffic volume, i.e., the number of vehicles of each of the $k$ groups passing through a fixed section of the selected section of the highway per unit of time in both directions in all traffic lanes, V/h;

$r_{vk,i}$ is a correction factor that takes into account the average speed of traffic flow ($v_k$, km/h) at the selected section of the road traffic system.

To determine the volume of emissions of pollutants from road transport in the area of regulated intersections, we use the formula

$$M_{Mi} = \frac{R}{40} \times \sum_{n=1}^{N_c} \sum_{k=1}^{N_g} \left( m_{ik}^Q \times Q_{max}^k \right), \tag{2}$$

where $R$ is the average duration of the prohibiting signal, min (including the intermediate cycle);

$N_c$ is the number of prohibiting strokes of the traffic light for a 20 min period of time, units;

$N_g$ is the number of vehicle groups;

$m_{ik}^Q$ is the specific emission of the $i$-th pollutant by vehicles of the $k$-th group, which are in the "queue" at the prohibiting traffic light, g/min;

$Q_{max}^k$ is the the length of the queue of vehicles of the $k$-th group, which are in the "queue" in the intersection zone at the end of the $n$-th cycle of the prohibiting traffic light, ed.

The model for assessing the emissions of pollutants from vehicles, taking into account the background of industrial enterprises, has the form

$$M_{em} = Lk_S \times \sum_i^k M_{k,i}^M \times G_k \times r_{vk,i} + \frac{R}{40} \times \sum_{n=1}^{N_c} \sum_{k=1}^{N_g} \left( m_{ik}^Q \times Q_{max}^k \right) + M_{pr,i}, \tag{3}$$

where $M_{pr,i}$—mass of pollutant emissions from industrial enterprises, gm/s.

Thus, to calculate the emissions from traffic flow, it is necessary to calculate the traffic intensity during the year on the basis of field surveys of the structure and intensity of the moving traffic flow on the main highways. Then, on the basis of the obtained data on the intensity, the calculation of vehicle emissions is obtained. With the help of the software product UPRZA Ecolog of the Integral company [57], dispersion maps for the main pollutants are built, on the basis of which points with the lowest concentrations of pollutants are allocated and the possibility of building parking lots on them is assessed.

## 4. Results

### 4.1. DSS for Parking Management

Decision-making on parking space organization is associated with the most appropriate options for the choice of specific conditions. For these purposes, it is rational to use modeling. Since the effective assessment of organizing parking lots for long-term vehicle storage (near residence places) and short-term parking (during the day) depends on different factors, the models used for the calculations will differ. It is possible to combine these models into a common management complex—for example, a decision support system (DSS) containing these models as an intelligent core.

Figure 1 shows the conceptual structure of the DSS, which will allow the consideration and comparison of options for possible solutions to improve the traffic situation in cities—in particular, solving the problem of parking management with changing traffic parameters and other factors.

The general management concept includes four processes:

1. The collection of initial data on the structure of the vehicle park, the objects on the road network and operational data from video cameras and other sensors from the infrastructure;

2. Organization of storage of collected initial data and secondary knowledge in the knowledge base gained as a result of the third process' execution;

3. Analysis of the received data, including the search for optimal solutions using the modeling and mathematical packages, assessment of the solution's effectiveness and their coordination with the requirements, including environmental;

4. Making decisions by specialists based on the knowledge gained.

Thus, the components of the decision support system correspond to these four stages. Their choice is based on the classical principle of feedback control, when the process of managing an object or system begins with the collection of information and the organization of its storage. Further, after the primary processing and analysis of the accumulated data, the decision-maker forms strategic and tactical decisions on the basis of the knowledge gained. The composition of the collection and storage components is determined by the trends in the development of the road transport infrastructure with a growing number of sensors and video surveillance cameras, data from which are accumulated in a shared data warehouse. The composition of the analysis unit is determined by the main tasks that are involved in organizing the parking space: choosing the location of the parking lot, organizing access to it and helping drivers to find free spaces.

### 4.2. Mathematical Model for Calculating Emissions from Vehicles and Industrial Facilities

Road transport requires more and more storage space due to the growth of motorization against the backdrop of an increase in the urban population. Parking lots are increasingly contributing to urban air pollution. Therefore, when organizing new parking lots, the most rational solution would be to take into account the general environmental situation in the city. We have carried out studies, on the basis of which we have constructed maps of dispersion of pollutants. The calculation was carried out on the basis of the above model: the traffic intensity was calculated on the basis of field studies of the structure and intensity of a moving traffic flow on the main highways of Naberezhnye Chelny, as well as emissions from enterprises. Figure 2 shows the map of the city from the Yandex maps service, overlaid with a map of CO dispersion, which has the greatest impact on the overall environmental situation.

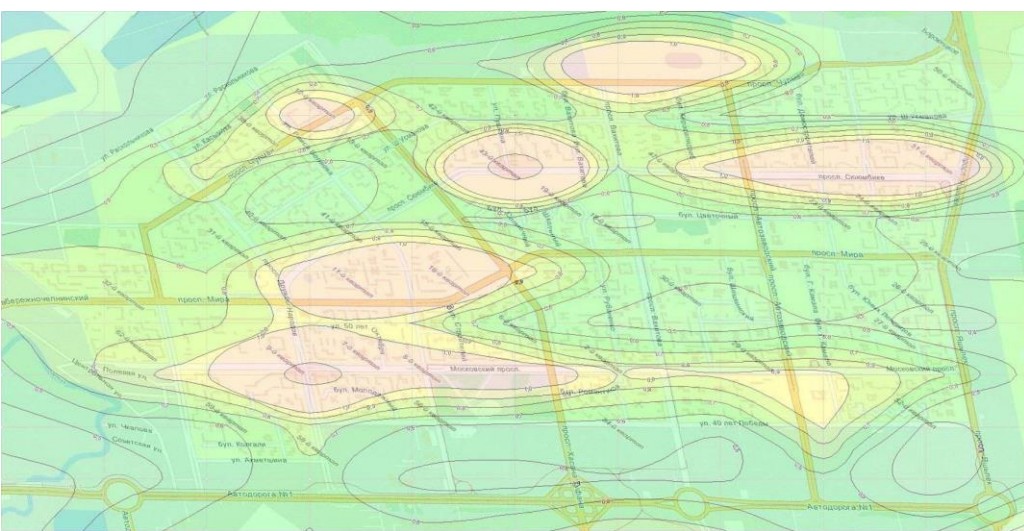

**Figure 2.** Map of CO dispersion in Naberezhnye Chelny.

Figure 3 shows the places where the construction of large parking lots will be the least dangerous for the environment due to the low concentration of CO in these areas. The source of the base map is also the Yandex maps service.

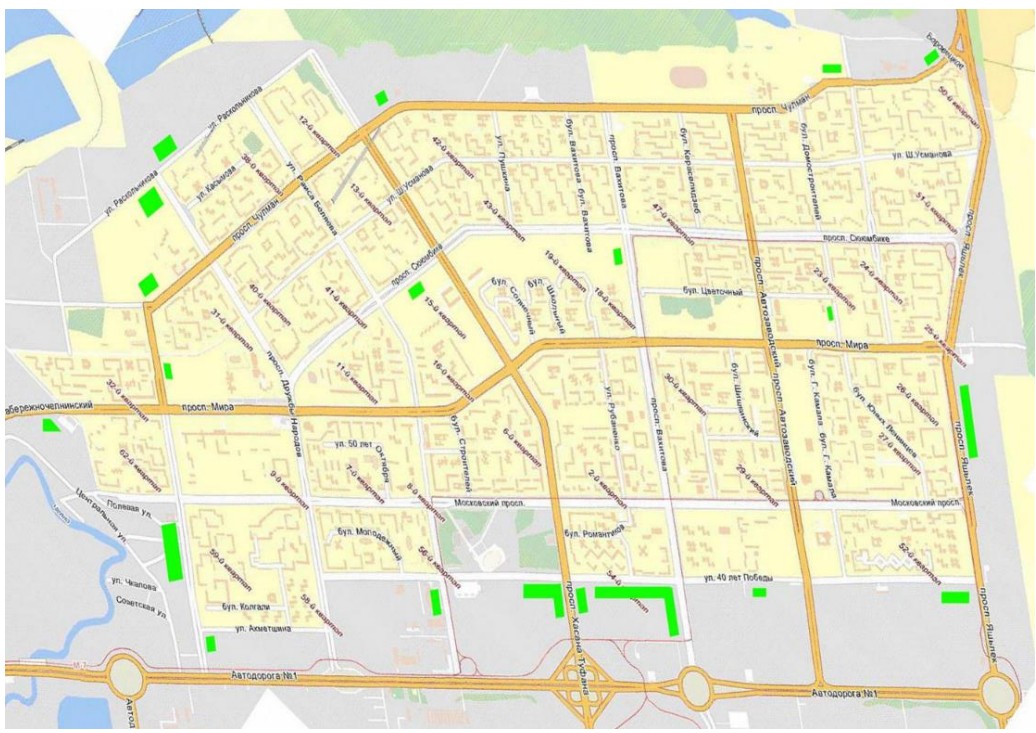

**Figure 3.** Places of possible construction of parking lots.

*4.3. Simulation of a Parking Lot near a Health and Sports Center*

Poorly organized parking lots near health and sports centers create serious problems in cities. An analysis of possible parking locations showed that, next to one of them, there is a fitness center with its own parking lot for customers (shown in Figure 3 with an asterisk). However, residents of nearby houses also call in and leave their vehicles for long-term storage. An analysis of the ecological situation of this location (Figure 2) showed the absence of a serious environmental load at present. However, according to the information provided by the fitness club management, it is planned to expand their activities and

increase the client base. In this case, there is a high risk of an increase in parking space occupancy, a lack of spaces for club customers, an increase in the time to find free spaces and, as a result, an increase in the environmental load on the territory.

The investigated object is the fitness center parking space. Currently, the entrance and exit from the parking lot are not controlled in any way and are not fixed. Residents of nearby houses and employees of shopping centers park their vehicles in this parking lot, often for the whole day, thereby taking away parking spaces from fitness club employees and clients. As a result, visitors randomly park their vehicles, sometimes causing accidents.

To solve the problem, the AnyLogic simulation environment was chosen. The AnyLogic traffic library allows one to simulate and visualize vehicle movement. The library supports detailed, highly efficient simulation of vehicle movement at the physical level. One can simulate the vehicle movement on the highway, street vehicle traffic, parking lots and any other system with vehicle, roads and lanes.

We built a model of the functioning of the fitness center parking space (Figure 4). As the initial data, the collected statistics of the intensity of the vehicle arrivals at the parking lot were used.

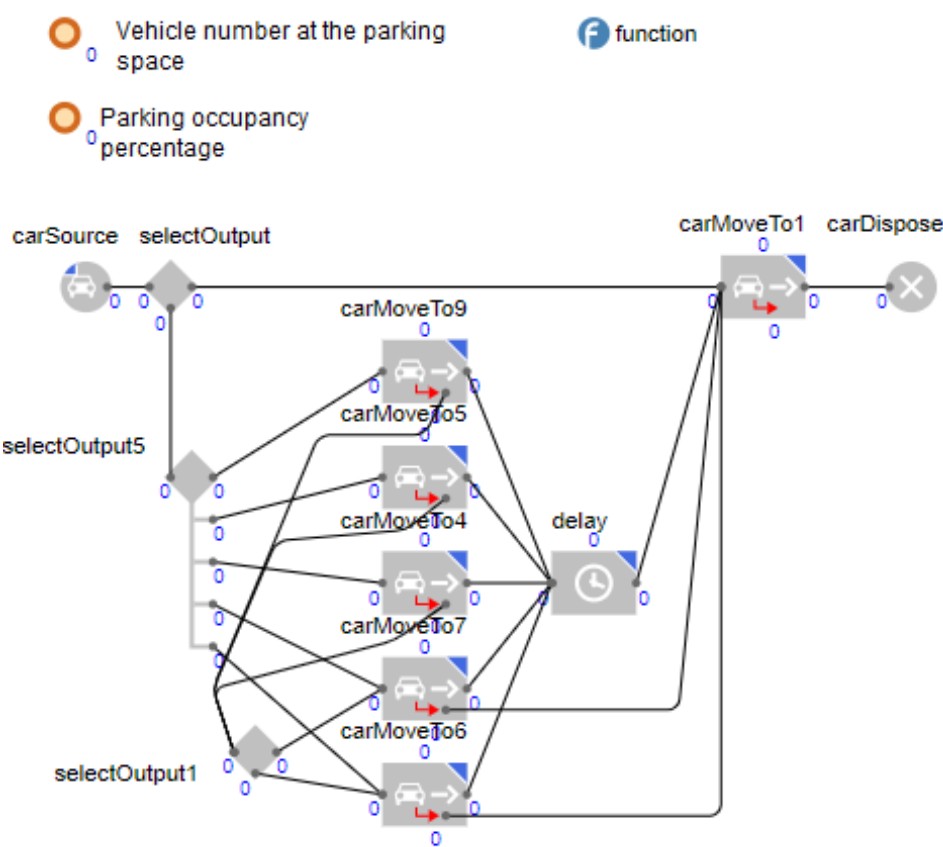

**Figure 4.** Model process diagram.

Model block parameters:
1. CarSource—creates a vehicle by fixing the agent on the roadway;
2. CarDispose—removes vehicle agents from the model, completing their cycle of action;
3. CarMoveTo—sets the direction of the vehicle agent movement, and calculates the path based on the specified direction;
4. SelectOutput—determines the movement of the vehicle on the road;
5. Delay—delays vehicles in the parking lot, simulating the time spent by a fitness center client in training.

In the simulation model, 5 parking areas are implemented in the entire parking lot. Thus, agents called vehicles are created in the carSource block. Next, a route is selected for these agents in the selectOutput block with a certain coefficient: to drive into the parking lot or drive past it. This is necessary because not all vehicles enter the parking area. Subsequently, agents that do not enter this territory follow the route on the map, pass by using the carMoveTo1 block and then are destroyed in the carDispose block. Otherwise, when the vehicles enter the parking area, the agents have a choice regarding which of the 5 parking lots to choose. This choice is not made by chance. After conducting a full-scale experiment in the parking space, the percentage of parking lots occupied by vehicles was identified and substituted into the simulation model. The carMoveTo (4,5,6,7,9) blocks are intended for the graphical installation of agents in the parking lot. Further, after installing agents in a parking space, a delay timer is needed to simulate the time for which the vehicle is left in the parking lot. After the time expires in the delay block, the vehicles leave the parking lot using the carMoveTo1 block, after which the agents are destroyed in the carDispose block.

Model visualization is presented in two forms: 2D (Figure 5) and 3D (Figure 6).

This model of the parking space, which does not record vehicles, was created in order to determine the degree of its maximum occupancy. The experiment performed on the simulation model of the original parking type (Figure 7) showed that, at a certain point in time, there are no parking spaces left and the fitness center clients cannot leave their vehicles in the parking lot.

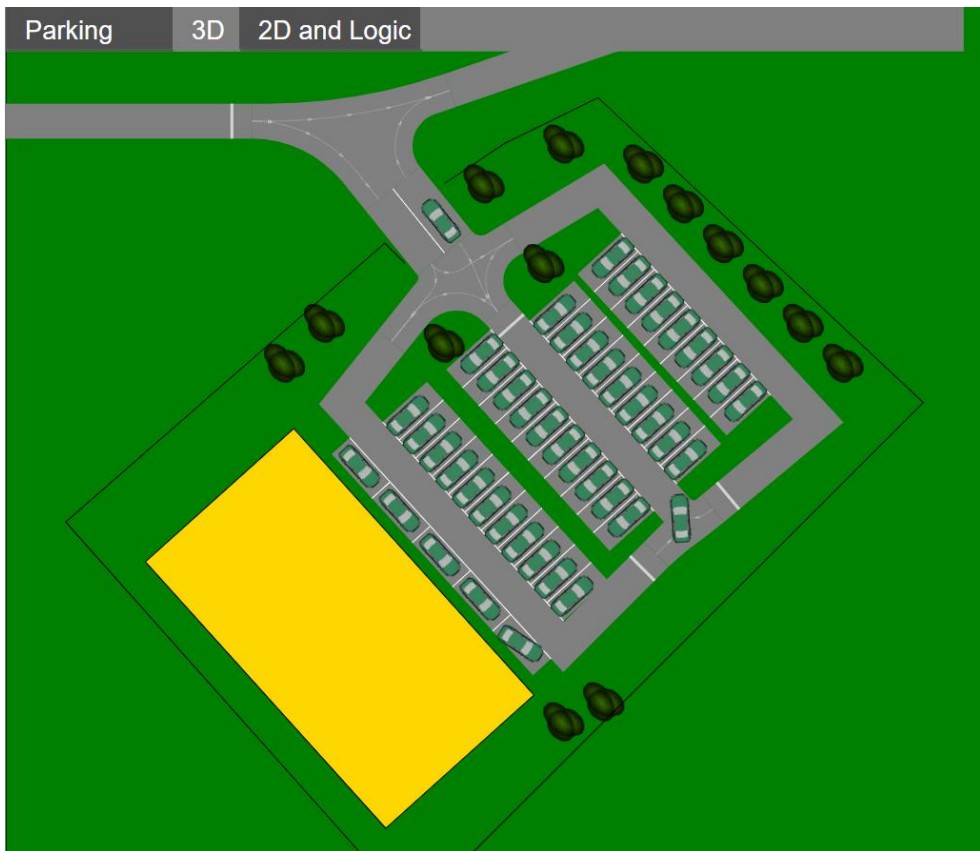

**Figure 5.** The 2D model animation.

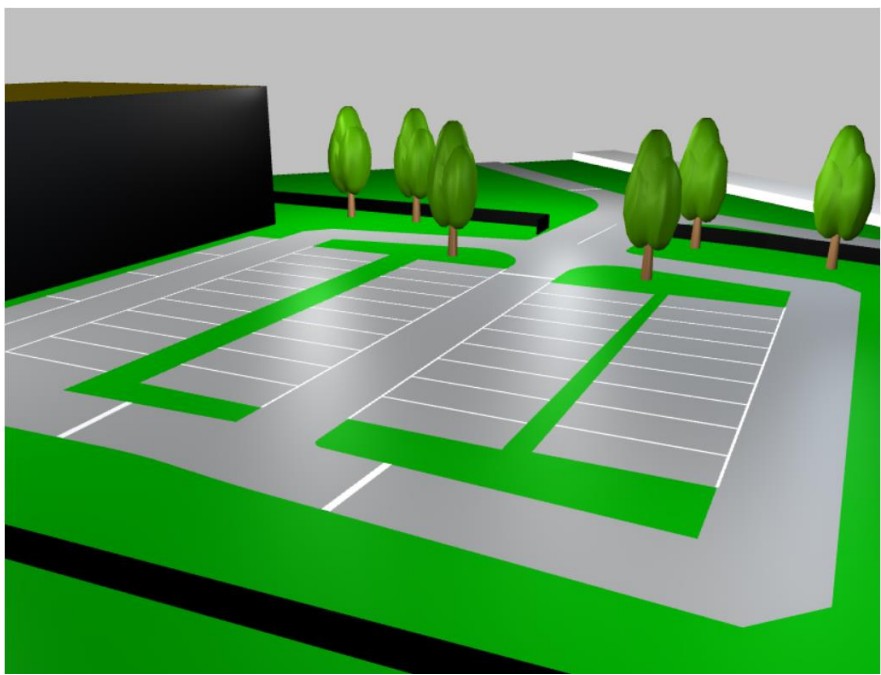

**Figure 6.** The 3D model animation.

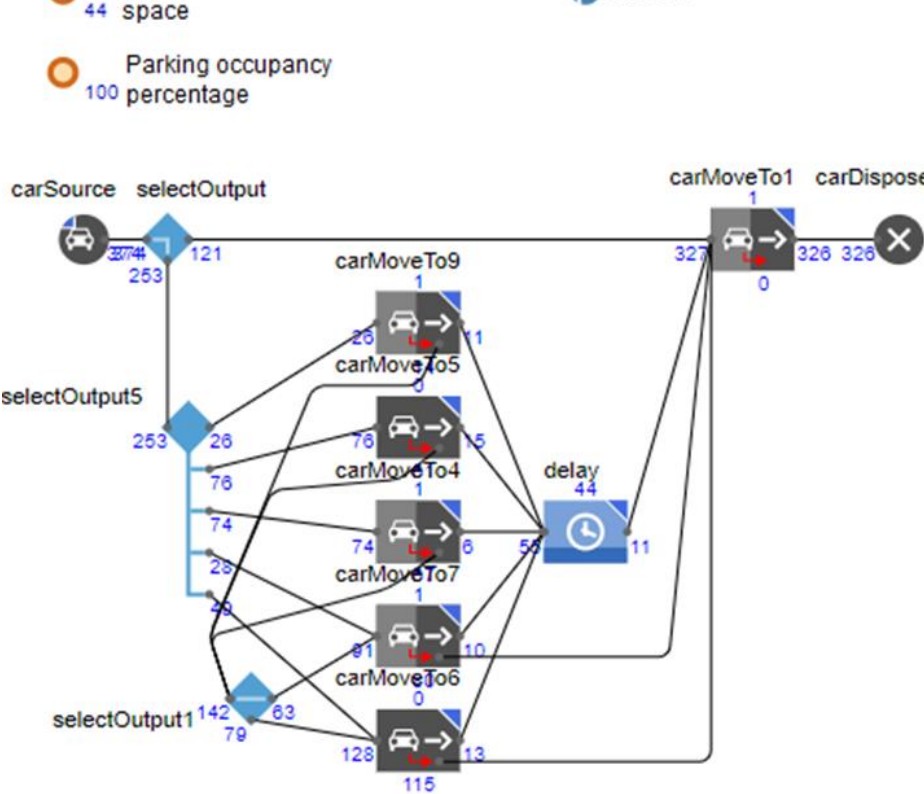

**Figure 7.** Simulation experiment.

To optimize the use of the parking space, we built a model including the control of entry and exit from the parking lot, where the pass required for entry and parking is the client card (Figures 8–10). These cards are differentiated: "black" cards are issued to those customers who can visit throughout the day, and "white" cards are issued to those who

visit only until 17.00. The ratio of these cards can change—for example, by conducting a marketing campaign: to redistribute customers, the fitness center can offer a good discount on services to the holders of "white" cards. Thus, by increasing the number of clients with "white" cards, evening hours can be unloaded. In the automated parking simulation model, the data on the vehicle intensity arrivals are taken from a classic relational database. Information about the time of vehicle arrivals at the parking lot is obtained from the database table in which arrivals are recorded; it is indicated in the carSource block.

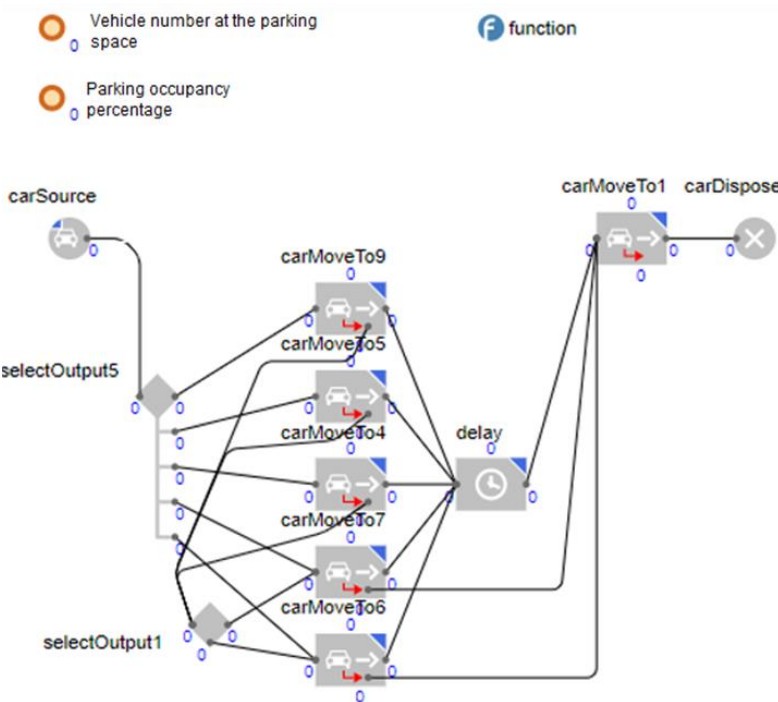

**Figure 8.** Process diagram of the new model.

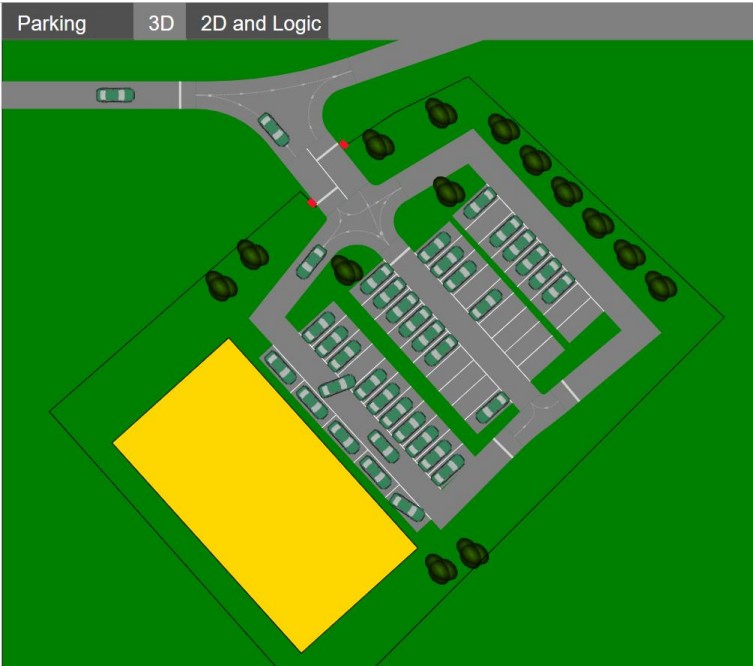

**Figure 9.** A 2D animation of the new model.

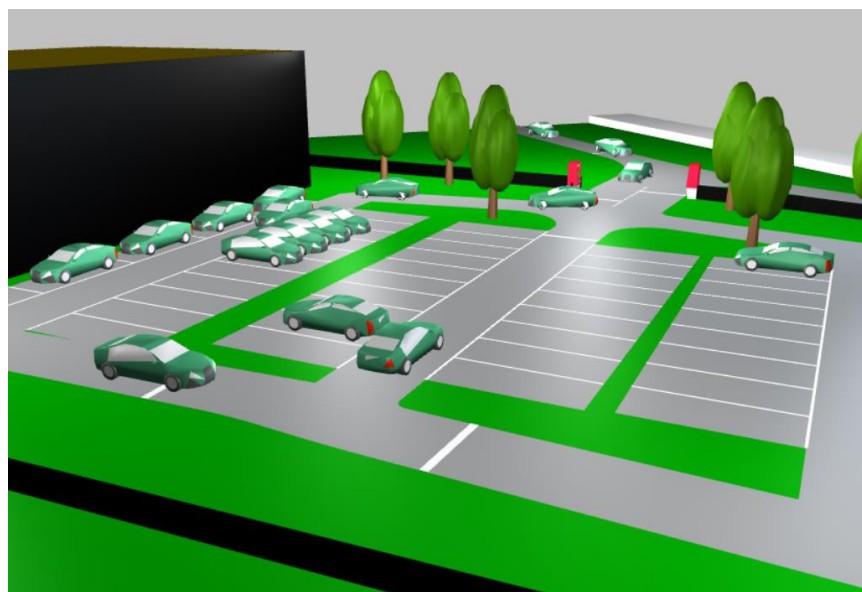

**Figure 10.** A 3D animation of the new model.

An experiment on an improved simulation model (Figure 11), with an implemented access control automation scheme using a client card, showed that the occupancy of the parking lot at the "peak" time did not exceed 75%.

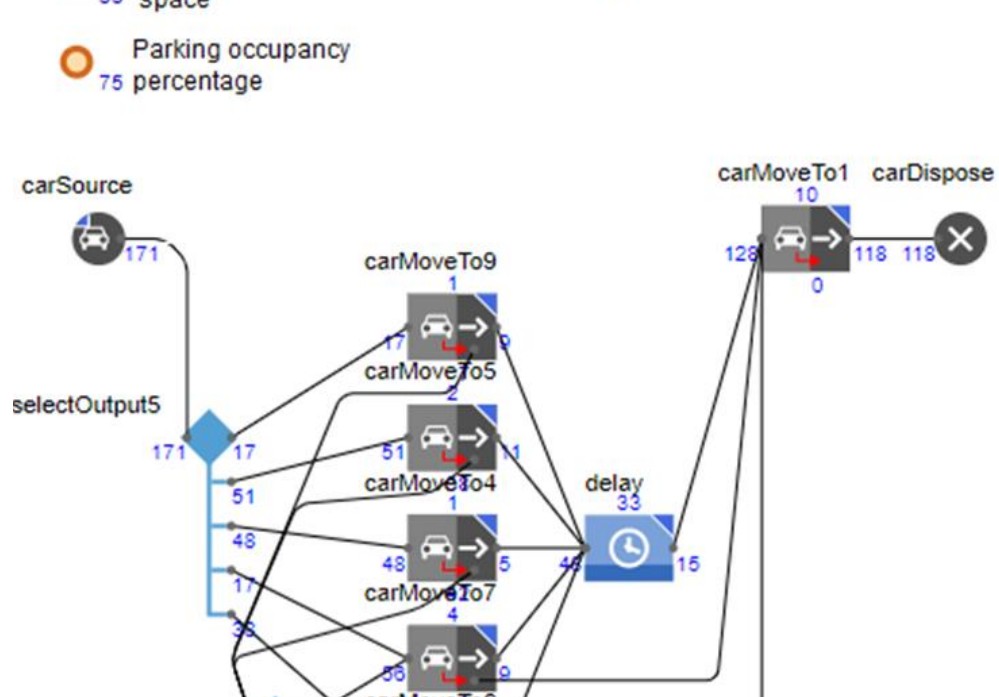

**Figure 11.** New simulation experiment.

Thanks to the simulation model, it is possible to analyze the parking lot occupancy on real data that are stored in the database of the fitness center information system.

The developed simulation model showed that automation of the parking space is necessary for the comfortable parking of employees and clients of the fitness center and will increase the parking capacity. Under the existing conditions, during peak hours, the parking lot was completely filled, and it was inconvenient for customers to visit by private vehicle. After automating the parking spaces, 25% of the spaces were vacated, which were previously occupied by employees and visitors of neighboring shopping centers and the Palace of Culture, as well as residents of nearby houses.

Thus, for solving such problems, simulation modeling is an excellent alternative, since it is more profitable to conduct a real experiment or manual calculation due to less labor and resource consumption required.

## 5. Discussion

In this work, we considered a methodology for searching for potential parking places in an urban area, taking into account the environmental situation. Based on the constructed maps of the dispersion of hazardous substances on the settlement territory, it is possible to identify parking spaces suitable for construction in order to prevent additional emissions in critical areas.

The prerequisites for this work were due to the specifics and characteristics of the considered city. Naberezhnye Chelny is a medium-sized town that is currently experiencing the trends of growing motorization and urbanization. This is reflected in the increasing density of the city center (the implementation of housing with tower blocks) and the increase in the area of the town (the building of new quarters on the outskirts of the town). The number of vehicles per person is also growing. The initially designed wide avenues and streets are still able to cope with the increase in traffic intensity. However, there is a problem of a lack of parking spaces for both long-term and short-term storage. The multi-level vehicle parks being built do not solve the problem, since the cost of a parking space in them is too high and sometimes exceeds the cost of the vehicle itself. In this case, the designers do not take into account the specifics of the average town, which is not a megalopolis, where the average income level is relatively low and does not allow residents to purchase a parking space next to a living space at a cost comparable to a quarter of that of the apartment itself. This forces vehicle owners to use the parking spaces of shopping, entertainment and sports centers located in close proximity to their places of residence. However, at the same time, potential and existing customers of these centers cannot use the parking spaces intended for them; they experience inconvenience and may refuse to visit them at all.

We demonstrated the possibility of optimizing parking spaces near cultural, sports and shopping centers using simulations. In this case, management is implemented through a decision support system. The considered example of modeling and optimizing parking near a fitness center is relevant for implementation in medium and small towns. An analysis of parking management research in cities did not reveal a similar example that demonstrates the solving of the identified problem and the implementation of a methodology to find locations and organize parking spaces, taking into account the environmental factor. The reviewed studies are aimed primarily at minimizing the time of using parking spaces with the help of payment, and optimizing routing in the process of finding a vacant space in a free parking lot. In this study, we made an assumption about the expediency of preventing a situation of congestion in the parking lot by organizing a complex logic of access to all interested parties within the framework of the necessary demand. This is possible by introducing a division of the flow of customers using customer cards: owners of "black" cards can enter the parking space and use the services throughout the entire day, while "white" card holders can visit only until 17.00. Accordingly, the cost of the subscription also varies. Of course, this complicates the control system and it is necessary to test the proposed logic by conducting experiments on a simulation model. Moreover, it is necessary to collect statistics on the use of parking spaces using recordings from video cameras or

by conducting field research. However, the conducted computer experiments prove the effectiveness of the proposed approach.

## 6. Conclusions

In today's market conditions, when tenants are more demanding and there is an excess supply, the high availability of office space with parking spaces is often the determining factor in the demand for properties. The organization of vehicle storage areas is currently a very topical issue in many cities. This is due to the increase in the growth rate of the global automotive sector. The current situation is forcing the municipal authorities and businesses to conduct a great deal of work to develop a comprehensive program for the organization and management of parking spaces.

The issue of parking space management is relevant, since, despite numerous studies in the optimization and intellectualization of the use of parking spaces, there are numerous unresolved issues. In particular, when deciding on the choice of location for organizing parking spaces, it is necessary to conduct a preliminary environmental analysis of the territory in order to identify places unfavorable for the construction of parking lots, already aggravated by the accumulation of pollutants. Further, when selecting potential places and organizing parking spaces, it is not enough to manage the search for free places based on intelligent systems. It is necessary to take into account the specifics of each particular settlement. Thus, for small and medium-sized towns with a low level of income among citizens, the organization of paid parking, as well as limits on free parking, may not only be ineffective but also destabilize the demand for the services of shopping, entertainment and sports centers. In this sense, the management of the demand for services itself and, accordingly, the division of visiting time and access to objects of attraction can be more effective. However, the task of the complex logic of access to the parking space requires preliminary approbation and the determination of parameters.

Good results can only be achieved with the help of complex solutions that combine advances in engineering and technology with methods and control tools. The relevance of using simulation models as the intelligent core of a DSS is obvious, since it will allow one to take into account the stochastic processes in the transport system. On the one hand, the agent-based approach makes it possible to set different levels of factors (speed, environmental class, emission level) for different models of road transport. On the other hand, using a discrete-event approach, one can, for example, consider a parking system near a shopping center as a queuing system, the lost profit amount by the owner of which can be estimated by counting the vehicles that did not find free parking spaces and traveled to other shopping centers, and then multiplying this amount by the average check size. We hope to consider this in further research.

The current study does not consider the further movement of vehicles, which, not being customers of the considered fitness center, are forced to continue to look for a parking space. Thus, we cannot assess how the introduction of the checkpoint at this facility affected the surrounding territories. Therefore, this parking lot should be considered in conjunction with neighboring sections of the city transport system and the road network. We plan to address this limitation in future works.

**Author Contributions:** Conceptualization, I.M.; methodology, I.M., V.M. and P.B.; formal analysis, I.M., A.B. and E.B.; investigation, I.M., D.S., V.M., P.B., A.B. and E.B.; resources, D.S. and V.M.; writing—original draft preparation, I.M., D.S., V.M., P.B. and A.B.; writing—review and editing, I.M., D.S., V.M., P.B., A.B. and E.B.; visualization, D.S. and V.M.; project administration, P.B. All authors have read and agreed to the published version of the manuscript.

**Funding:** The reported study was funded by RFBR, project number 19-29-06008\21.

**Data Availability Statement:** Data can be obtained from the corresponding auth.

**Conflicts of Interest:** The authors declare no conflict of interest.

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
