# Peer review of "Rational Organization of Urban Parking Using Microsimulation"

_infrastructures, doi:10.3390/infrastructures7100140_

Round 1

Reviewer 1 Report

Interesting article on the current problem of road traffic engineering, which is parking. Recently, this is another article in this field that I am reviewing.

The literature review needs to be corrected. Currently, there is one paragraph for each item. At the end of chapters 3.1., 3.2. and 3.3. there should be a summary of what the authors read along with a conclusion (what is similar, what are the differences). These are currently unrelated paragraphs.

In the title of chapter 3.2. I suggest changing the word "free" to, for example, "vacant", it is not about "bесплатные", but about "свободное".

Line 119 et seq. The causes of road congestion are, to a greater extent, the lack of prioritization of the road network (roads have both transit and local functions) as well as the traffic volume close to the capacity.

Line 183 - The number 2 should be subscript.

There is no separate "Methods" section in the article. After the literature review, the method should be presented and only then "results", "discussion" and "conclusions".

Line 412 - no drawing source.

Line 430 - k in the variable name should be subscript. "Интенсивность движения" should be translated as "traffic volume", not "traffic intensity", the unit is vehicle per hour [V/h]

Line 448 requires an English translation.

Please standardize lines 457 (CO) and 460 (pollutants).

Please explain the dangers of verse 462. I do not understand this fragment of the article.

In which program were the analysis and drawing 2 made? Please describe it in the "Methods" section with appropriate references to the literature describing this program.

Please complete the source of the base map for Figures 2 and 3.

Please describe the traffic simulation method in the AnyLogic software (in "Methods" section).

Figure 4 contains descriptions in Russian. I understand them, but not every reader knows Russian language. Please translate and correct the drawing. The same remark applies to Figures 7 and 11.

Explain the concept of "agent" - line 501.

The summary describes that the method is based on field studies. However, there is no description of these studies in the content of the article. Please describe them and information whether they were carried out - before or after the beginning of the war in Ukraine (if this event affects the traffic at the place where the research is carried out).

Author Response

Dear reviewer,

We appreciate your  feedback on our paper. Please see file which explain our responses to the comments, and we highlighted all changes in the manuscript as track changes.

Best regard,

Authors

Reviewer 2 Report

The article Rational Organization of Urban Parking Using Simulation attracts very interesting content with its abstract. A presentation of solutions to the problem of parking in small and medium-sized cities is promised here. This sounds like a very useful research goal with high potential for future real-world application. The filling of cities with cars is becoming critical, and the process of finding free parking spaces and distributing them to users could be very beneficial.
But the article itself does not provide answers to crucial questions and basically does not offer any exciting solutions. In Chapter 2, an extensive review of the procedures used in the literature to solve the problem of finding and distributing free parking spaces is carried out. Unfortunately, this chapter has no substantial overlap with the authors' contributions themselves. From the text of the article, I understood that the authors solved a complex mathematical problem of parking space distribution using AnyLogic software. I don't see anything wrong with that, but the boundary conditions of the solved problem need to be clearly defined. In this case, no solution conditions are set at a general level. A specific case of an urban district with intensive development and a lack of parking spaces is presented here. Unfortunately, even this problem is not solved in the end, and the authors resort to solving the very narrow issue of filling the parking lot at the fitness center, which is potentially filled both by visitors to the nearby shopping center and residents from adjacent properties. The proposed solution using an entrance barrier and client card for clients of the fitness center definitely does not need simulation and does not solve the problem of optimal selection of parking spaces. It would be more interesting to deal with at least the entire street, block of buildings and the way of distribution of free parking spaces.
Here I have to return to the research and the introduction of the solution itself. One of the critical tasks that must be solved during the distribution of free parking spaces is the acquisition of source data, i.e., the current occupancy of parking spaces and the demand of users (motorists) for an available parking space. Neither issue is mentioned. Furthermore, nowadays, I would welcome the integration of V2X (V2V) technology, which has the potential to provide quality source data at a minimal cost.
The presented idea certainly has potential, but I would welcome a more complex solution and a more extensive application example.
A note on the text: On line 448, the text remained in Russian.
A question on equations 1-3: How do you plan to implement electric cars into your calculation?
A general question: How do you want to implement users with specific requirements? For example, electric car users want to stay near the charging station, handicapped persons have priority places, some inhabitors have reservation cards, etc.

Author Response

(The authors gave the same response as above.)

Reviewer 3 Report

The article studies the possibility of using simulation models to find rational options for organizing parking spaces and further use such models in decision support systems (DSS) as an intelligent core. It is an interesting topic looking for a solution to a quite critical and relevant problem, however, it needs to be reorganized considerably and several sections adjusted. The points for improvement are detailed below:

1. The title should include more information about the type of simulation performed

2. The writing of the Abstract should be improved. For example, when mentioning "The first one impedes the 13 movement of vehicles, lead to congestion and traffic jams. The second one makes it difficult for 14 direct customers to access services." It is not understood what it refers to, since the 2 points are far back in the text. It is not convenient to include the opinion of the authors in the abstract, when you mention: "In our opinion, it is the most adequate option for an intelligent city parking space management". This statement should be supported by the bibliographic review performed. I think it is important to define whether the article is a review of the state of the art or research, in order to focus the content of the article, not to combine the two types of document.

3. The text of the introduction should extend a few more paragraphs. It should include a broader Background, supported by more references. In addition, you must clearly specify what is the problem to be solved with the proposal. It must also include what is the novelty that the proposal introduces and finally the method used.

4. The "Materials and Methods" section should also be much more detailed. Most of the text included in section 4.2 should go in this "Materials and Methods" section, since part of this content does not present results, it presents the mathematical support used.

5. Section 3 presents a bibliographic review of topics related to the topic of the article, however, although several articles are presented, adequate conclusions are not made in each subsection, nor is a general conclusion. I think it is very important to make conclusions in each subsection (3.1, 3.2, 3.3, and 3.4) and a general conclusion, which explains the differences between what is currently found and the article's proposal.

6. In section 4.1 the conceptual structure of the proposed DSS is presented, but there is no explanation or justification of where said proposal comes from. Nor is it explained how the sources were obtained, nor the procedure used to make the graphs in figures 2 and 3.

7. The discussion of the results obtained should be improved, explaining the relevance of the results and the parameters and procedures to be considered, in order to apply the proposal.

8. With the changes proposed in the previous sections, the proposed conclusions should be improved.

Author Response

(The authors gave the same response as above.)

Round 2

Reviewer 2 Report

Dear  authors,
Thank you for the cover letter where you answer my questions in detail. The actual version of your paper is more understandable in aims and development strategies.
The idea of supporting parking lots' dimensions with simulation is good, but the realization is still poor. My personal feeling about the article doesn't change. I don't understand the final solution with cards for fitness center clients. Such a solution doesn't need simulation. I supposed you are planning to support some overall solution of parking in the city in the range, for example, one block. Moreover, I hoped you interact with cars or drivers to recommend the optimal place to stay. Nothing from these two aims is realized, although you promised it in the abstract and the introduction.
I recommend you change the abstract and introduction and describe your aims in truth. 

Author Response

Dear reviewer,

thank you for the time spent on reviewing our manuscript and the valuable comments on its improvement. Please see the attachment

Reviewer 3 Report

Regarding the corrections made by the authors, I have the following observations, which I consider important to take into account, before approving it for publication: 

-        It is mentioned that corrections were made in the abstract, but these changes were not properly identified in the last version of the article (I think it is necessary to indicate with red text, what was modified). It was necessary to review all the text again, in order to verify that it had been changed.

-        I consider that the introduction still requires improvement and expansion. Regarding the changes identified (with red text) they are minimal. The recommendations made were not accepted. In addition, a final paragraph indicating how the other sections of the article are organized is recommended.

-        The recommendations requested in the previous Section 3 were not even followed. In the last review the following was requested: “I think it is very important to make conclusions in each subsection (3.1, 3.2, 3.3, and 3.4) and a general conclusion, which explains the differences between what is currently found and the article's proposal”.

-        Regarding point 4.1, each proposed component is detailed, but it is not justified why said component is proposed.

-        The discussion and conclusions sections require considerable improvement. The discussion should be much broader and consider many more relevant aspects.

Author Response

Dear reviewer,

thank you for the time spent on reviewing our manuscript and the valuable comments on its improvement. Please see the attachment.

Round 3

Reviewer 2 Report

I haven't any other comments on the paper.

Reviewer 3 Report

The authors have made the requested adjustments. It is recommended to review the wording, there is still room for improvement.